energy/organic chemistry/computer modelling and simulation

superadiabtic combustion, porous media, excess enthalpy function

**Author for correspondence:**
Mingming Mao
e-mail: maomm2019@163.com

This article has been edited by the Royal Society of Chemistry, including the commissioning, peer review process and editorial aspects up to the point of acceptance.

# Theoretical analysis of superadiabatic combustion for non-stationary filtration combustion by excess enthalpy function

Junrui Shi, Mingming Mao, Yongqi Liu and Jinsheng Lv

School of Transportation and Vehicle Engineering, Shandong University of Technology, Zibo, 255049 Shandong, People's Republic of China

MM, 0000-0003-2372-8451

The superadiabatic combustion for non-stationary filtration combustion is analytically studied. The non-dimensional excess enthalpy function ($H$) equation is theoretically derived based on a one-dimensional, two-temperature model. In contrast to the $H$ equation for the stationary filtration combustion, a new term, which takes into account the effect of non-dimensional combustion wave speed, is included in the $H$ equation for transient filtration combustion. The governing equations with boundary conditions are solved by commercial software Fluent. The predictions show that the maximum non-dimensional gas and solid temperatures in the flame zone are greater than 3 for equivalence ratio of 0.15. An examination of the four source terms in the $H$ equation indicates that the thermal conductivity ratio ($\Gamma_s$) between the solid and gas phases is the dominant one among the four terms and basically determines $H$ distribution. For lean premixed combustion in porous media, the superadiabatic combustion effect is more pronounced for the lower $\Gamma_s$.

## 1. Introduction

Excess enthalpy combustion in porous media [1,2], also known as energy concentration, draws constant interest from researchers due to its wide range of applications and outstanding features of heat recovery and pollutant control. The energy concentration phenomenon and combustion characteristics in porous burners have been investigated extensively in the literature including experimental [3–8], numerical [9–16] and analytical studies [16–27]. Babkin *et al.* [21] presented a detailed review of this issue and stated that the phenomenon of energy concentration is

more widespread in nature than was assumed previously. As for gas combustion in inert porous media, the mechanism of superadiabatic combustion both for travelling and stabilization flames has been studied.

Heat recuperation by solid-phase conduction and radiation was attributed as the main mechanism of superadiabatic combustion [6–10]. Through this recirculation, it was theoretically possible to achieve a local temperature higher than the adiabatic flame temperature of the mixture entering the burner. This was called the superadiabatic combustion. Based on the one-dimensional flame theory, the analysis by Min & Shin [5] showed that the heat is recirculated to the unburned mixture both by conduction and radiation of the solid phase. Lee *et al.* [6] theoretically and experimentally studied laminar premixed flames stabilized inside a honeycomb ceramic burner. They found that in the reaction region the radiation is comparable to the conduction in the solid phase. Drayton *et al.* [7] experimentally studied syngas production from superadiabatic combustion of an ultra-rich methane–air mixture. Experimental results showed that the reciprocal flow burner, due to its high heat recuperation efficiency, can support self-sustained combustion of ultra-rich methane–air mixtures up to an equivalence ratio of eight, well beyond the conventional flammability limit associated with a methane–air flame in free space. Bedoya *et al.* [8] studied the effects of pressure and air/fuel equivalence ratio on the combustion characteristics of three different porous burners, employing a volume-averaged one-dimensional model and three-dimensional direct pore level simulation (DPLS) on real geometries of sponge-like structures. Their results showed that three-dimensional DPLS can predict the burning velocity values at high pressure, while the one-dimensional model yields lower values than the experiments.

For transient gas combustion in porous media, there exist two relevant parameters, namely, a nearly constant combustion wave speed and a maximum temperature, as demonstrated in references [3,16,17]. Zhdanok *et al.* [3] first revealed the coupling between the thermal and combustion waves. According to their formulation, when the heat loss to the surroundings was ignored, the maximum non-dimensional superadabatic combustion temperature $\theta_{g,\max}$ became a function of the ratio of the combustion wave speed $u_w$ and thermal wave speed $u_t$ in the porous medium:

$$\theta_{g,\max} = 1 - \frac{u_w}{u_t}. \tag{1.1}$$

Gas combustion in homogeneous porous media typically is a transient process and the flame propagates in the same or opposite direction as the flow of inlet mixtures, depending on the equivalence ratio and the mixture velocity. The case $u_w/u_t = 1$ corresponds to the most pronounced superadiabatic effect. Equation (1.1) also indicates that the case $u_w = 0$ separates two mechanisms, namely superadiabatic and subadiabatic combustion mechanisms. Here $u_w > 0$ and $u_w < 0$ correspond to the superadiabatic and subadiabatic combustion, respectively. Shi *et al.* [16] demonstrated analytically that the mechanism of superadiabatic combustion is a result of the overlap of the thermal and combustion waves under certain conditions. To extend the lean flammability limit, reciprocating superadiabatic combustion of premixed gases in inert porous media was proposed. In this system, the direction of gas flow was periodically changed at a regular interval, thus the heat stored in the burner was used for fully preheating the fresh mixture. Through this method, the effect of superadiabatic combustion was realized and the maximum combustion temperature in the porous media was about four times that of the adiabatic combustion of the inlet mixture [4].

In contrast to the pronounced superadiabatic combustion effect for the transient combustion in porous media, where the flammability limit can be extended to an extremely low equivalence ratio, only a slight superadiabatic combustion effect was observed for stable combustion in porous media, a lean flammability limit of about 0.4 was reported for natural gas [22]. Both solid conduction and radiation were the dominant models of heat recirculation. Wichman & Vance [19] analysed laminar premixed flame annihilation using a thin flame asymptotic method. They calculated an excess enthalpy function along the flame and the effect of Lewis number on the excess enthalpy was discussed. Coutinho *et al.* [14] conducted a one-dimensional numerical simulation on premixed combustion in two-layer porous media burner. They found that increasing the ratio between the solid thermal conductivity to that of the gas phase of the preheating section reduced peak temperatures in the combustion region. Sahraoui & Kaviany [9] conducted a numerical study of the flame structure and flame propagation speed in a structured porous medium. In their model, the radiant exchange between the solid surfaces is ignored. They studied the effects of the arrangement of porous media, the conductivity ratio between the solid phase to the gas phase, pore size and flame position on the flame speed and superadiabatic combustion. The results showed that a 60% increase in the flame

speed compared to the speed in free space and a slight superadiabatic combustion effect under the condition of the equivalence ratio is close to unity.

Pereira's research group [22–25] carried out a series of studies on laminar stationary premixed flames within porous inert media. Their analysis was based on the excess enthalpy function applied to a set of one-dimensional volume-averaged equations. They extended their model in their studies to different conditions. The extended model allowed the construction of an analytical solution valid over a large range of equivalence ratios. In their experimental and analytical studies on excess enthalpy in a two-layer burner [22], it was shown that the $H$ is a function of modified Lewis number, the ratio of the solid and the gas-phase effective conductivities and the porosity of the porous media. Subsequently, Pereira *et al.* [23] presented an analytical solution for the structure of premixed flames in porous media using the asymptotic expansion method. In addition, Pereira *et al.* [25] developed an extended model which is valid for lean mixtures with the condition of intermediate values of the interphase heat transfer between the gas and solid. Based on this model, the lean flammability limit and the maximum superadiabatic temperature were obtained.

Recently, Vahid & Chanwoo [28] investigated analytically superadiabatic combustion for lean premixed gas with equivalence ratio range from 0.55 to 1.0. They presented a complete set of the closed-form solutions for the temperature profiles of the solid and gas phases, and the fuel concentration profile. The effects of the inlet gas velocity, fuel equivalence ratio, porosity, etc., on combustion temperatures were discussed.

As mentioned above, the mechanism of superadiabatic combustion is revealed from different aspects including thermal and combustion waves coupling, heat recirculation and RSCP. However, theoretical analysis of the mechanism of superadiabatic combustion for non-stationary filtration combustion using $H$ is not yet available. To extend the lean flammability limit, one may always expect to realize superadiabatic combustion and to get a maximum combustion temperature under certain conditions, but the predominant factor that determines the superadiabatic combustion temperature is still not clear. These questions are addressed in this paper. Jiang *et al.* [29,30] modelled premixed combustion in a randomly packed bed, and found that premixed flames are concentrated in the thin reaction zone.

In the following section, we formulate a mathematical model of filtration combustion in a packed bed and the $H$ equation is theoretically derived based on a one-dimensional and two-temperature model. Subsequently, the source terms in the non-dimensional $H$ function is examined and the effects of $\Gamma_s$ and porosities on the superadiabatic combustion effect are discussed.

# 2. Mathematical formulation

## 2.1. Problem formulation

An inert porous media burner reported in Zhdanok *et al.* [3] was considered in this study. Figure 1 shows the schematic diagram of the porous burner. The experimental apparatus consists of a quartz tube with an internal diameter of 76 mm and is filled with 5.6 mm solid alumina spheres. The wall was insulated to reduce the heat loss to the surroundings.

## 2.2. Governing equation

In a previous study, Pereira *et al.* [22] presented a detailed analysis of stationary premixed combustion in an inert porous media burner. In this work, we consider the laminar transient filtration combustion of a lean mixture, where the combustion wave propagates in the same direction as inflow of the gaseous mixture.

For simplicity, we introduce the following assumptions:

(1) The working gas is non-radiating and the gas flow in the porous medium is laminar.
(2) The pressure loss in the burner is neglected.
(3) The effective thermal conductivity of the solid includes solid conduction and radiation.
(4) Heat loss through the burner walls to the surroundings is neglected.
(5) All solid thermophysical properties are constant.

Under the above assumptions, a set of differential equations can be obtained and expressed as follows [17].

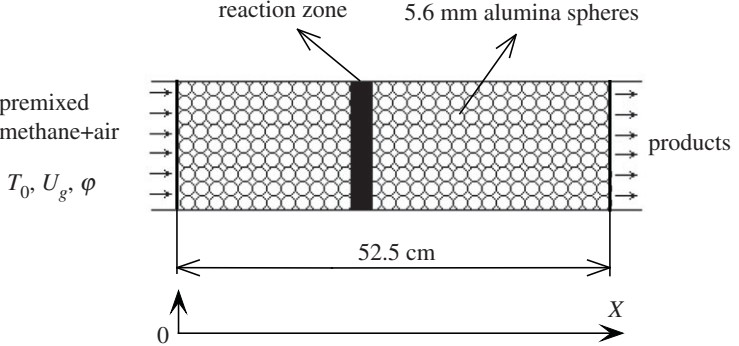

**Figure 1.** Schematic diagram of porous media burner.

**Table 1.** The main symbols.

| Nomenclature | |
| --- | --- |
| $c$ specific heat, kJ m$^{-1}$ K$^{-1}$ | $h_v$ convective heat transfer between solid and gas phases, W m$^{-3}$ K$^{-1}$ |
| $T$ temperature, K | $u_g$ gas mixture velocity, m s$^{-1}$ |
| $t$ time, s | $T_{ad}$ adiabatic combustion temperature, K |
| $T_0$ ambient temperature, K | $u_{g,in}$ mixture velocity at inlet, m s$^{-1}$ |
| $u_w$ combustion wave speed, m s$^{-1}$ | $u_t$ thermal wave speed, m s$^{-1}$ |
| $x$ horizontal coordinate, m | $Y$ mass fraction |
| non-dimensional variables | |
| $\theta$ temperature | $y$ mass fraction |
| $H$ enthalpy function | |
| criterion number | |
| $Le_s$ modified Lewis number | $Le$ Lewis number |
| greek symbols | |
| $\lambda$ thermal conductivity, W m$^{-1}$ K$^{-1}$ | $\lambda_{eff}$ effective thermal conductivity, W m$^{-1}$ K$^{-1}$ |
| $\rho$ density, kg m$^{-3}$ | $\varepsilon$ porosity |
| $\dot{\omega}$ reaction rate, K mol m$^{-3}$ s$^{-1}$ | $\Gamma_s$ conductivity ratio between the solid and gas phases |
| g gas | s solid |

*Species conservation equation*

$$\varepsilon \rho_g \frac{dY}{dt} - \varepsilon \rho_g D \frac{d^2 Y}{dx^2} + \varepsilon \rho_g u_g \frac{dY}{dx} + \varepsilon \dot{\omega} = 0. \tag{2.1}$$

*Gas-phase energy equation*

$$\varepsilon \rho_g c_g \frac{dT_g}{dt} + \varepsilon \rho_g c_g u_g \frac{dT_g}{dx} = \varepsilon \lambda_g \frac{d^2 T_g}{dx^2} + h_v (T_s - T_g) + \varepsilon Q \dot{\omega}. \tag{2.2}$$

*Solid-phase energy equation*

$$(1 - \varepsilon) \rho_s c_s \frac{dT_s}{dt} = \lambda_{eff} (1 - \varepsilon) \frac{d^2 T_s}{dx^2} + h_v (T_g - T_s). \tag{2.3}$$

For convenience, the main symbols used in this paper are listed in table 1. Here $\rho_g$ and $u_g$ are the gas mixture density and velocity, respectively; $Y$ is fuel mass fraction; $\varepsilon$ is porosity of the packed bed. The reaction rate $\dot{\omega}$ [17] is considered to obey the first-order Arrhenius equation and is expressed as

$\dot{\omega} = \rho_g Y A_0 \exp(-E_a/RT_g)$. $T_g$ and $T_s$, are gas and solid temperatures, respectively; $\lambda_g$ is the gas thermal conductivity; $h_v$ is convective factor between the gas and solid phases; $Q$ is the low heating value of fuel; $\lambda_{\text{eff}}$ is effective thermal conductivity of the porous media.

## 2.3. Construction of $H$ equation

In a previous study, Pereira *et al.* [22] have constructed the $H$ equation for stationary premixed combustion in a porous medium. However, in our model, the system of equations describes the transient combustion wave propagation in porous burner, hence, it differs from that of Pereira *et al.* [22]. To distinctly clarify the problem and for latter analysis, the construction process is performed in the following section.

For a fully developed wave moving at a constant speed $u_w$ in the packed bed, equations (2.1)–(2.3) can be rewritten in a reference frame of a new coordinate $X = x - u_w t$, which is attached to the reaction front moving in the $x$-direction with the combustion wave velocity $u_w$. Noting that $u_g \gg u_w$, these equations are transformed as

$$-\varepsilon \rho_g D \frac{d^2 Y}{dX^2} + \varepsilon \rho_g u_g \frac{dY}{dX} + \varepsilon \dot{\omega} = 0, \tag{2.4}$$

$$\varepsilon \rho_g c_g u_g \frac{dT_g}{dX} = \varepsilon \lambda_g \frac{d^2 T_g}{dX^2} + h_v(T_s - T_g) + \varepsilon Q \dot{\omega}, \tag{2.5}$$

and

$$-(1-\varepsilon)\rho_s c_s u_w \frac{dT_s}{dX} = \lambda_{\text{eff}}(1-\varepsilon)\frac{d^2 T_s}{dX^2} + h_v(T_g - T_s). \tag{2.6}$$

The following non-dimensional parameters and variables are introduced [17]:

$$Le_s = Le\Gamma_s, \; Le = \frac{\lambda_g}{\rho_g c_g D}, \; \Gamma_s = \frac{\lambda_{\text{eff}}}{\lambda_g}, \; u = \frac{u_w}{u_t}, \; y = \frac{Y}{Y_{F,n}},$$

$$\theta_g = \frac{T_g - T_0}{T_{\text{ad}} - T_0}, \; \theta_s = \frac{T_s - T_0}{T_{\text{ad}} - T_0} \; \zeta = \frac{\varepsilon \rho_g c_g u_g}{\lambda_{\text{eff}}} X,$$

and

$$W = y \exp\left[\frac{-\beta(1-\theta_g)}{1-\alpha(1-\theta_g)}\right], \; N = \frac{\lambda_{\text{eff}} h_v}{(\varepsilon \rho_g u_g c_g)^2}, \; Da = \frac{\rho_g A_0 e^{-\beta/\alpha}\lambda_{\text{eff}}}{(\varepsilon \rho_g u_g)^2 c_g},$$

where $y$, $\theta_g$ and $\theta_s$ are non-dimensional fuel mass fraction, gas and solid temperatures, respectively. $T_{\text{ad}}$ and $T_n$ are the gas adiabatic combustion temperature and ambient temperature, respectively; $Le_s$ is the modified Lewis number; $\Gamma_s$ is the thermal conductivity ratio between the solid and gas phases; $u$ is the non-dimensional combustion wave speed; $Y_{F,n}$ is the fuel mass fraction at the burner inlet. Thermal wave velocity $u_t$ is determined by [17]

$$u_t = \frac{\varepsilon \rho_g c_g u_g}{(1-\varepsilon)\rho_s c_s}. \tag{2.7}$$

Substituting these parameters and variables into equations (2.4)–(2.6), we find

$$\varepsilon \frac{dy}{d\zeta} = \varepsilon \left(\frac{1}{Le_s}\right)\frac{d^2 y}{d\zeta^2} - \varepsilon Da W, \tag{2.8}$$

$$\varepsilon \frac{d\theta_g}{d\zeta} = \varepsilon \left(\frac{1}{\Gamma_s}\right)\frac{d^2 \theta_g}{d\zeta^2} + N(\theta_s - \theta_g) + \varepsilon Da W, \tag{2.9}$$

$$-u \frac{d\theta_s}{d\zeta} = (1-\varepsilon)\frac{d^2 \theta_s}{d\zeta^2} - N(\theta_s - \theta_g). \tag{2.10}$$

Combining equation (2.8) with equations (2.9) and (2.10), we obtain

$$\frac{d(y + \theta_g)}{d\zeta} = \frac{d^2}{d\zeta^2}\left[\frac{1}{Le_s}y + \frac{1}{\Gamma_s}\theta_g + \left(\frac{1}{\varepsilon} - 1\right)\theta_s\right] + u \frac{d\theta_s}{d\zeta}. \tag{2.11}$$

A new variable $H$, called the enthalpy function, is introduced as [19]

$$H = y + \theta_g - 1. \tag{2.12}$$

The physical significance of $H$ is that it essentially defines the total enthalpy of the gas, including the thermal and chemical enthalpies. From the definition of $H$, we find:

$$\frac{dH}{d\zeta} = \frac{d^2H}{d\zeta^2} + \left(\frac{1}{Le_s} - 1\right)\frac{d^2y}{d\zeta^2} + \left(\frac{1}{\Gamma_s} - 1\right)\frac{d^2\theta_g}{d\zeta^2} + \left(\frac{1}{\varepsilon} - 1\right)\frac{d^2\theta_s}{d\zeta^2} + u\frac{d\theta_s}{d\zeta}. \tag{2.13}$$

For simplicity the second, third, fourth and fifth terms in the right-hand side of equation (2.13) are replaced with $\Delta, \Psi, \Pi$ and $\Theta$. We find,

$$\frac{dH}{d\zeta} = \frac{d^2H}{d\zeta^2} + \Delta + \Psi + \Pi + \Theta. \tag{2.14}$$

The above equation shows that the enthalpy function is controlled by the combined effect of the modified Lewis number, the thermal conductivity ratio between the solid and gas phases, the porosity of packed bed and the non-dimensional combustion wave speed. The fifth term $\Theta$ in the right side of equation (2.14) is a new term compared to the $H$ equation induced from the stable combustion in porous media [22]. In the following section, we assume that Lewis number equals to unity, hence we obtain the expression $Le_s = \Gamma_s$, which means that the influence $Le_s$ on $H$ is same as that of $\Gamma_s$.

## 2.4. Analysis of $H$ equation and solution

Equation (2.14) is reduced to the following expression when $\varepsilon = 1$

$$\frac{dH}{d\zeta} = \frac{d^2H}{d\zeta^2} + \left(\frac{1}{Le} - 1\right)\frac{d^2y}{d\zeta^2}. \tag{2.15}$$

The above equation is the enthalpy function for laminar free flame and coincides with the result obtained in [19]. When $Le = 1$, the source term in equation (2.15) vanishes, thus $H$ is equal to zero everywhere in the flame. In other words, there is no excess or defect of enthalpy in the flame zone.

When $u = 0$, one can rewrite equation (2.14) as

$$\frac{dH}{d\zeta} = \frac{d^2H}{d\zeta^2} + \Delta + \Psi + \Pi. \tag{2.16}$$

The above equation is the $H$ equation for stationary premixed combustion in a porous medium when the effect of $u$ on $H$ vanishes. The terms $\Delta, \Psi$ and $\Pi$ terms in equation (2.14) coincide with those of ref. [22]. Their meaning has been discussed by Pereira et al. [22]. The fifth term in the right side of equation (2.14) is taken into account in this study due to the presence of the travelling combustion wave.

The last four terms in equation (2.14) can be considered as positive or negative source terms depending on the signs for the diffusion-like terms. The conductivity of the solid is always greater than that of gas mixture and, therefore, $Le_s > 1$ everywhere for $\zeta$. Thus, $\Delta$ represents the axial enhanced diffusion effect on the reactant distribution by the prefactor of $Le_s^{-1} - 1$ and this leads to an increase in the flame thickness. The term $\Psi$ describes the effect of solid conductivity on the non-dimensional gas temperature. As $\Gamma_s$ is always greater than 1, the prefactor $1/\Gamma_s - 1$ is always negative and this term will cause an excess or defect enthalpy along $\zeta$. Since $\varepsilon$ is smaller than one, the prefactor $1/\varepsilon - 1$ is always positive, thus $\Pi$ causes an excess of the enthalpy. The term $\Theta$ accounts for the effect of $u$ on the non-dimensional solid temperature distribution. In this study for $CH_4$/air in a range of equivalence ratios from 0.15 to 0.45, $u$ is always positive, and this term causes an excess enthalpy from the preheat region to the maximum $\theta_s$ and a small defect enthalpy after this point. The combined effects of the mentioned four terms determine the $H$ distribution.

## 2.5. Boundary conditions

At the inlet $y = 0, \theta_g = 0, u_g = u_{g,in}$ is imposed in the computation. At the outlet, we assume that the flow is fully developed, which means that all of the gradients of the non-dimensional variables equal zero.

## 2.6. Initial conditions and solution

A uniform square grid with a size of 1 mm was used in the computation domain. The initial $\theta_s$ is assumed to be the same as the initial pre-heating temperature profile reported in Zhdanok et al. [3].

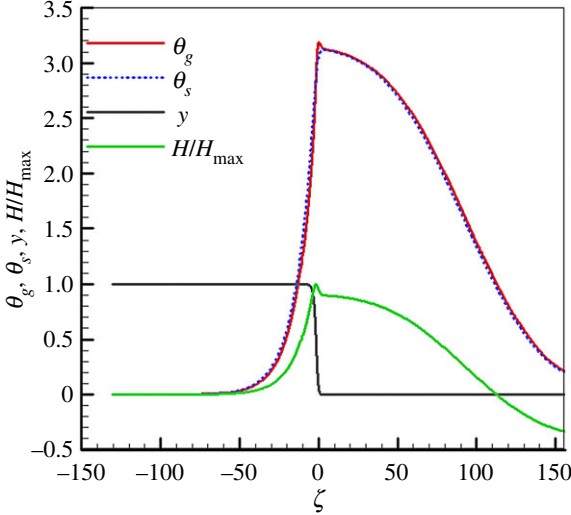

**Figure 2.** $\theta_g$, $\theta_s$, $y$ and $H/H_{\mathrm{max}}$ distributions ($\varphi = 0.15$, $u_{g,\mathrm{in}} = 0.43$ m s$^{-1}$, $\Gamma_s = 20$).

**Table 2.** Thermodynamic properties and coefficient of reaction.

| | | |
|---|---|---|
| porosity | $\varepsilon$ | 0.4 |
| solid density (kg m$^{-3}$) | $\rho_s$ | 1300 |
| specific heat of solid (J kg$^{-1}$ K$^{-1}$) | $c_s$ | 2500 |
| ambient temperature (K) | $T_0$ | 298 |
| activation energy (J mol$^{-1}$) | $E_a$ | 141 000 |
| frequency factor (1 s$^{-q}$) | $A_0$ | 2.2E8 |
| universal gas constant (J mol$^{-1}$ K$^{-1}$) | $R$ | 8.314 |
| heat exchange coefficient (W m$^{-3}$ K$^{-1}$) | $H_v$ | 80 000 |
| heat of reaction (J kg$^{-1}$) | $Q$ | 50 144 000 |

The $H$ equation is based on the solution of equations (2.8)–(2.10) using the finite-volume method by Fluent. The gas properties are approximated by the air properties at the adiabatic combustion temperature of the inlet CH$_4$/air mixture. Properties used for the computation are shown in table 2.

# 3. Results and discussion

## 3.1. $\theta_g$, $\theta_s$, $y$ and $H/H_{\mathrm{max}}$ distribution

Note that the non-dimensional gas temperature is defined as $\theta_g = (T_g - T_n/T_{\mathrm{ad}} - T_n)$. It is clear that there is both superadiabatic and subadiabatic combustion, which occurs when $\theta_g$ is greater than or less than one, respectively. Thus, superadiabatic combustion takes place when $\theta_g > 1$, in which the maximum gas temperature is greater than the adiabatic flame temperature. Otherwise, the superadiabatic effect is no longer realizable. The greater the $\theta_g$, the more pronounced superadiabatic combustion effect.

Figure 2 shows the non-dimensional gas and solid temperatures, fuel mass fraction and normalized excess enthalpy ($H/H_{\mathrm{max}}$) distributions under the experimental conditions of Zhdanok *et al.* [3] for $\varphi = 0.15$, $u_{g,\mathrm{in}} = 0.43$ m s$^{-1}$ and $\Gamma_s = 20$. The entire flame zone is divided into a pre-heat zone, reaction zone and a post-flame zone. The location $\zeta = 0$ corresponds to the reaction zone.

As shown in figure 2, a high and wide non-dimensional temperature zone for both gas and solid phases is observed. The maximum $\theta_g$ and $\theta_s$ in the flame zone are greater than 3. This indicates that the superadiabatic combustion effect is pronounced. A lean mixture ($\varphi = 0.15$) cannot support a flame in an open tube. However, in our case the combustion is stable and the predicted results coincide with those of Zhdanok *et al.* [3], who investigated filtration combustion in a packed bed. A stable

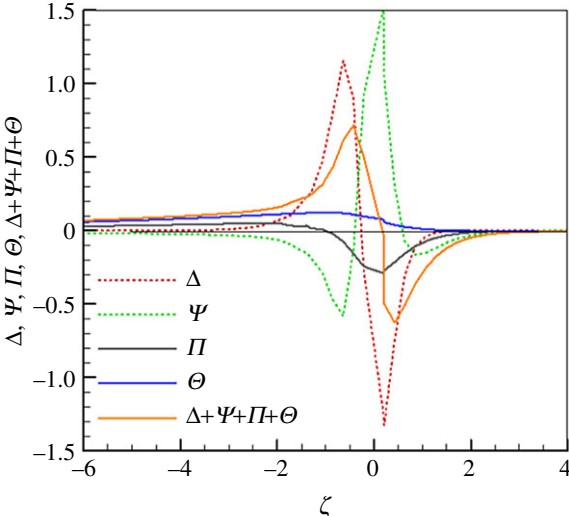

**Figure 3.** Variation of source terms in equation (2.14) along flame ($\varphi = 0.25$, $u_{g,\text{in}} = 0.43$ m s$^{-1}$, $\Gamma_s = 60$).

combustion wave was confirmed experimentally for methane/air mixtures with an equivalence ratio of 0.15. The influence of $\Gamma_s$ on the maximum $\theta_g$ will be discussed later.

As seen in figure 2, in the pre-heat zone no fuel has yet been consumed and the gas mixture is effectively preheated by convection. Consequently, $H$ increases in this region and reaches its maximum in the reaction zone. Then $H$ decreases when the fuel is consumed and the gas temperature decreases in the post-flame zone. Figure 2 shows that the $H$ is positive in the pre-heat and reaction zones and is negative at the outlet.

## 3.2. $\Delta, \Psi, \Pi, \Theta$ and $\Delta + \Psi + \Pi + \Theta$ distributions

We examine the relative magnitudes of the different terms of the source terms in the $H$ equation along the flame for $\varphi = 0.25$, $u_{g,\text{in}} = 0.43$ m s$^{-1}$ and $\Gamma_s = 60$. Figure 3 shows that $\Pi$ and $\Theta$ are relatively small and negligible in the whole flame region. The term $\Theta$ is positive in the preheat zone and decreases to zero in the post-flame zone, whereas $\Pi$ is positive in the pre-heat zone and decreases to a minimum near the flame zone. It then increases in the post-flame zone and approaches to zero along $\zeta$. From figure 3, $\Delta$ and $\Psi$ dominate the source term in the flame zone, although they show the reversed trends. In the pre-heat zone, $\Delta$ is positive and increases because fuel has to be consumed, whereas $\Psi$ is negative and decreases due to the increase in gas temperature. In the flame region, $\Delta$ and $\Psi$ reach their minimum and maximum values, respectively. In the post-flame zone, $\Delta$ increases while $\Psi$ decreases, and then they all approach rapidly to zero along $\zeta$. As observed in fig. 3, the total source term $\Delta + \Psi + \Pi + \Theta$ mainly depends on the distributions of $\Delta$ and $\Psi$.

We recall that $\Theta$ is a new term when compared to the $H$ equation for stationary filtration combustion. However, from figure 3 we can see that $\Theta$ is positive in the pre-heat zone, which is beneficial for superadiabatic combustion. However, its value is relatively small among the four source terms of the $H$ equation and it contributes little to the source term under this condition. To further clarify the relative magnitudes of different source terms in equation (2.14), more computations are conducted, but we present here only one result because they show the same trends. As shown in figure 4, the relative magnitudes of $\Delta, \Psi, \Pi, \Theta$ and $\Delta + \Psi + \Pi + \Theta$ are very similar to those in figure 3. We may conclude that $\Theta$ does benefit the superadiabatic combustion, but it is an insignificant term among the four source terms, the terms $\Delta$ and $\Psi$ play an important role in determining the magnitude of the source term in the $H$ equation.

## 3.3. Influence of $\Gamma_s$ on $\theta_{g,\text{max}}$

The effect of $\Gamma_s$ on $\theta_{g,\text{max}}$ is shown in figure 5, in which the theoretical predictions [25] and experimental results [3] are also presented. Our predictions show that increasing $\Gamma_s$ leads to a decrease in $\theta_{g,\text{max}}$ due to the effective thermal recirculation by solid matrix through the high-temperature zone. That is, the

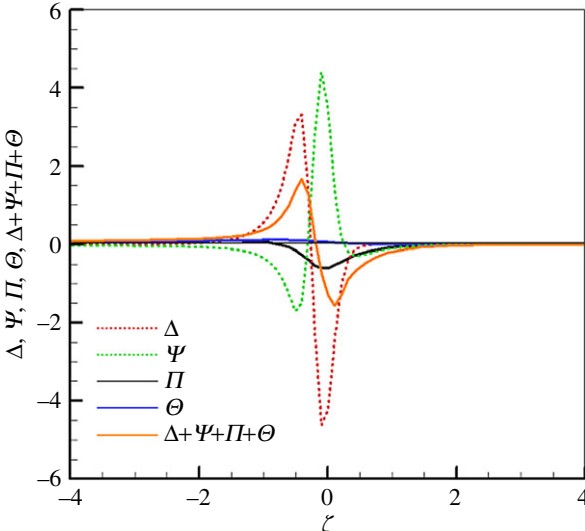

**Figure 4.** Variation of source terms in equation (2.14) along flame ($\varphi = 0.35$, $u_{g,\text{in}} = 0.86$ m s$^{-1}$, $\Gamma_s = 80$).

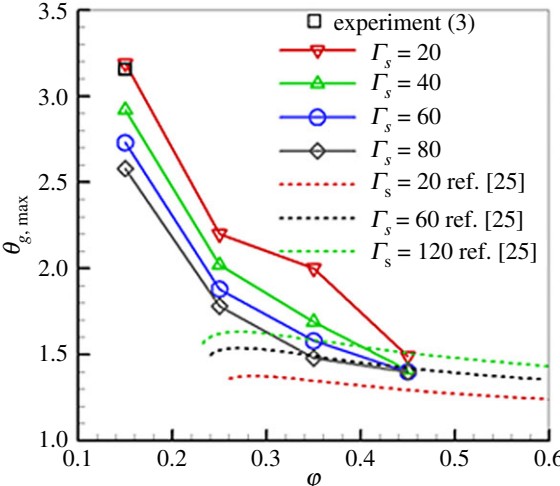

**Figure 5.** Effect of $\Gamma_s$ on $\theta_{g,\text{max}}$ ($u_{g,\text{in}} = 0.43$ m s$^{-1}$).

superadiabatic effect is amplified when $\Gamma_s$ is decreased. For extremely lean combustion, the relatively smaller heat recirculation generates a higher combustion zone. Thus, it is advantageous to use a porous medium with a lower heat conductivity in the burner to produce excess enthalpy combustion for the lean premixtures. Furthermore, a good agreement between the experimental results and our predictions for $\Gamma_s = 20$ is observed. However, $\theta_{g,\text{max}}$ decreases as $\Gamma_s$ increases.

As demonstrated in figure 5, the predicted effect of $\Gamma_s$ on $\theta_{g,\text{max}}$ and by Pereira *et al.* [25] show an opposite trend. Pereira *et al.* [25] showed that increasing $\Gamma_s$ leads to an increase in $\theta_{g,\text{max}}$. They concluded that leaner mixtures become flammable as $\Gamma_s$ was increased. This discrepancy might be attributed to the different $u_{g,\text{in}}$ used in the studies. The present $\theta_{g,\text{max}}$ are obtained under conditions of the fixed $u_{g,\text{in}}$ (0.43 m s$^{-1}$) for all $\Gamma_s$. By contrast, Pereira *et al.* [25] analysed stationary flames, where the laminar flame speed is equal to $u_{g,\text{in}}$. They used an expression for the flame speed from the previous work [23]. From this expression, we can see that the flame speed is a function of $\varphi$. In other words, for the same $\Gamma_s$, in their predictions different $u_{g,\text{in}}$ might be used with variation of $\varphi$. For stationary filtration combustion, the combustion wave has to be confined and held inside the combustor, thus different $u_{g,\text{in}}$ are used with a variation of $\varphi$. However, as illustrated by many research works [3,16], $u_{g,\text{in}}$ has a significant influence on superadiabatic combustion in porous media. In addition, different porous media used (foam ceramic) in the studies might be another contributing factor to the discrepancy. Further research on this subject is required.

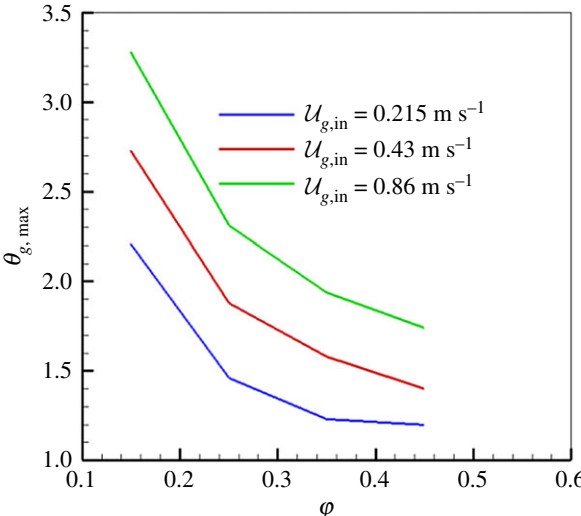

**Figure 6.** Effect of $u_{g,in}$ on $\theta_{g,max}$ ($\Gamma_s = 60$).

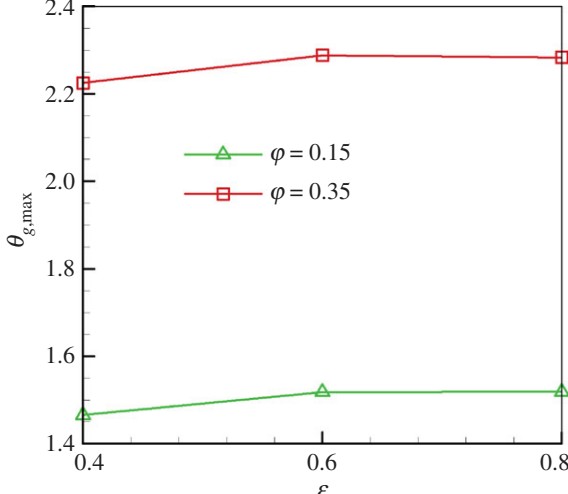

**Figure 7.** Effect of $\varepsilon$ on $\theta_{g,max}$ ($u_{g,in} = 0.83$ m s$^{-1}$, $\Gamma_s = 80$).

## 3.4. Influence of $u_{g,in}$ on $\theta_{g,max}$

The effect of $u_{g,in}$ on $\theta_{g,max}$ is shown in figure 6 for $\Gamma_s = 60$ for different $\varphi$. As depicted in figure 6, $u_{g,in}$ has remarkable influence on $\theta_{g,max}$ and the superadiabatic combustion effect becomes pronounced when $u_{g,in}$ increases for the fixed $\varphi$. This is because more fuel is fed into the system as $u_{g,in}$ increases, thus more heat is stored in the porous medium due to the greater specific heat of solid compared with that of gas phase. This indicates that an increased $u_{g,in}$ leads to an extended lean flammability limit. For extra-lean mixture combustion in porous media, the burner performance can be improved by increasing $u_{g,in}$. Moreover, as shown in figure 6, the superadiabatic effect becomes weaker as $\varphi$ increases for fixed $u_{g,in}$. Our predictions are in agreement with the predictions of Shi *et al.* [16] and Henneke *et al.* [31], who conducted numerical studies on the experiments of Zhdanok *et al.* [3]. They showed that the superadiabatic combustion effect gradually becomes less obvious as $\varphi$ approaches about $\varphi = 0.5$ for CH$_4$/air combustion in porous burner. At the same time, the combustion speed approaches zero, which separates the superadiabatic and subadiabatic combustion mechanisms.

## 3.5. Influence of porosity on $\theta_{g,max}$

Figure 7 demonstrates the effect of the porosity on $\theta_{g,max}$ for $\Gamma_s = 80$ and $u_{g,in} = 0.86$ m s$^{-1}$. We can see that the porosity has a weak influence on the $\theta_{g,max}$. Increasing porosity from 0.4 to 0.8 induces a slight

increase in $\theta_{g,\max}$. Results indicate that the superadiabatic combustion effect is more pronounced for greater matrix porosities. It is noted that, when the porosity varies for the packed bed, the properties of the porous media, diffusion processes in the burner and the heat transfer between the gas and solid phases change accordingly. However, in this work we assume that all of the solid thermophysical properties are constant, the effect of porosity on the solid thermal conductivity and $h_v$ are ignored, and this result is just a qualitative approximation. However, our results on the effect of porosity on $\theta_{g,\max}$ show an opposite trend to the predictions by Pereira et al. [25]. In the equivalence ratio range of $0.4 < \varphi < 0.8$, their results showed that $\theta_{g,\max}$ linearly decreased when porosity increased. Again, the discrepancy between the predicted trends may be attributed to the different $u_{g,in}$ used in the studies stated above.

# 4. Conclusion

Based on a one-dimensional and two-temperature model, the excess enthalpy equation for the premixed combustion in porous media burner has been theoretically derived. The governing equations with boundary conditions are solved by commercial software Fluent. It is shown that the non-dimensional enthalpy equation is controlled by the combined effect of the modified Lewis number, the thermal conductivity ratio between the solid and gas phases, the porosity of the packed bed and the non-dimensional combustion wave speed. When $Le = 1$, the modified Lewis number equals the thermal conductivity ratio between the solid and gas phases. Our results show the thermal conductivity ratio between the solid and gas phases has a significant influence on $\theta_{g,\max}$ and the sum of the source terms in the $H$ equation. In addition, calculated results show that the $\Delta$ and $\Psi$ terms dominate the source term in the $H$ equation and basically determine the $H$ distribution in the flame zone. Furthermore, the superadiabatic combustion effect is more pronounced for a lower Lewis number under the condition of lean premixed combustion in porous media.

Data accessibility. All data and code are available from the Dryad Digital Repository: https://doi.org/10.5061/dryad.1vhhmgqq8 [32].

Authors' contributions. J.S. and M.M. set up the model, Y.L. and J.L. wrote the manuscript. All authors gave final approval for publication.

Competing interests. We declare we have no competing interests.

Funding. This work is supported by the National Natural Science Foundation of China (grant nos. 51876107).

Acknowledgements. The author is grateful to many colleagues with whom he had the privilege to interact and collaborate over the years and whose work is partially referenced in this article.

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
