## [Reviewer comments · Royal Society Open Science]

Review History

RSOS-201038.R0 (Original submission)

Review form: Reviewer 1

Is the manuscript scientifically sound in its present form?

Yes

Are the interpretations and conclusions justified by the results?

Yes

Is the language acceptable?

Yes

Do you have any ethical concerns with this paper?

No

Have you any concerns about statistical analyses in this paper?

No

Recommendation?

Accept as is

Comments to the Author(s)

The manuscript (RSOS-201038) presents a theoretical analysis of superadiabatic combustion for the non-stationary filtration combustion in porous medium. The excess enthalpy function (H equation) is derived based on the energy equations and species conservation equation. An analysis of the terms in H equation is conducted and it is found that the thermal conductivity ratio between the solid and gas phases is a dominative factor among the four terms. It is an interesting research on the filtration combustion in porous medium which will help to understand the mechanism of the superadiabatic combustion. My specific comments are as follows,

1. More new and relevant literature should be cited and introduced in the introduction.
2. How about the grid of computing domain?
3. The deviation in the Figure 5 is attributed to the different structure of porous media applied in the experiment. More discussions are needed to clarify the problem.
4. The conclusion and discussion should be more detailed.
5. It is recommended to carefully check the grammar and English language before resubmission.

To list but a few as below:

Line 14, page 9, "the thermal conductivity ratio between the solid and gas phases (Γ s) is" should be "the thermal conductivity ratio (Γ s) between the solid and gas phases is".

Line 12, page 2, "using superadiabatic combustion" should be "from superadiabatic combustion".

Line 7 page 3, "opposite directions" should be "opposite direction".

Line 9 page 3, "Equation. (1)" should be "Equation (1)".

Line 17, page 5, "H equation are" should be "H equation is".

T0, Tn are confused in the manuscript.

Line 16, page 10, "is imposed" should be "are imposed".

Review form: Reviewer 2

Is the manuscript scientifically sound in its present form?

No

Are the interpretations and conclusions justified by the results?

Yes

Is the language acceptable?

No

Do you have any ethical concerns with this paper?

No

Have you any concerns about statistical analyses in this paper?

No

Recommendation?

Major revision is needed (please make suggestions in comments)

Comments to the Author(s)

1. English should be improved.
2. In the Abstract, you should clearly mention whether your paper is experimental or analytical and what software was used.
3. The most important quantitative results should be mentioned in the abstract.
4. The novelty of the paper should be highlighted.
5. A list of abbreviations and acronyms should be added.

6. It is not a good idea that the authors refer a simple sentence to several references.
7. All equations need an appropriate reference.
8. The objectives of the work should be clearly highlighted.
9. What software was used in this study?
10. How the results were validated?

Decision letter (RSOS-201038.R0)

Dear Dr Mao:

Title: Theoretical analysis of superadiabatic combustion for non-stationary filtration combustion by excess enthalpy function
Manuscript ID: RSOS-201038

The editor assigned to your manuscript has now received comments from reviewers. We would like you to revise your paper in accordance with the referee and Subject Editor suggestions which can be found below (not including confidential reports to the Editor). Please note this decision does not guarantee eventual acceptance.

Please submit your revised paper before 02-Sep-2020. Please note that the revision deadline will expire at 00.00am on this date. If we do not hear from you within this time then it will be assumed that the paper has been withdrawn. In exceptional circumstances, extensions may be possible if agreed with the Editorial Office in advance. We do not allow multiple rounds of revision so we urge you to make every effort to fully address all of the comments at this stage. If deemed necessary by the Editors, your manuscript will be sent back to one or more of the original reviewers for assessment. If the original reviewers are not available we may invite new reviewers.

On behalf of the Subject Editor Professor Anthony Stace and the Associate Editor Dr Debashree Ghosh.

RSC Associate Editor:
 Comments to the Author:
 (There are no comments.)

RSC Subject Editor:
 Comments to the Author:
 (There are no comments.)

Reviewers' Comments to Author:
 Reviewer: 1

Comments to the Author(s)

The manuscript (RSOS-201038) presents a theoretical analysis of superadiabatic combustion for the non-stationary filtration combustion in porous medium. The excess enthalpy function (H equation) is derived based on the energy equations and species conservation equation. An analysis of the terms in H equation is conducted and it is found that the thermal conductivity ratio between the solid and gas phases is a dominative factor among the four terms. It is an interesting research on the filtration combustion in porous medium which will help to understand the mechanism of the superadiabatic combustion. My specific comments are as follows,

1. More new and relevant literature should be cited and introduced in the introduction.
2. How about the grid of computing domain?
3. The deviation in the Figure 5 is attributed to the different structure of porous media applied in the experiment. More discussions are needed to clarify the problem.
4. The conclusion and discussion should be more detailed.
5. It is recommended to carefully check the grammar and English language before resubmission.

To list but a few as below:

Line 14, page 9, "the thermal conductivity ratio between the solid and gas phases (Γ s) is" should be "the thermal conductivity ratio (Γ s) between the solid and gas phases is".

Line 12, page 2, "using superadiabatic combustion" should be "from superadiabatic combustion".

Line 7 page 3, "opposite directions" should be "opposite direction".

Line 9 page 3, "Equation. (1)" should be "Equation (1)".

Line 17, page 5, "H equation are" should be "H equation is".

T0, Tn are confused in the manuscript.

Line 16, page 10, "is imposed" should be "are imposed".

Reviewer: 2

Comments to the Author(s)

1. English should be improved.
2. In the Abstract, you should clearly mention whether your paper is experimental or analytical and what software was used.

3. The most important quantitative results should be mentioned in the abstract.
4. The novelty of the paper should be highlighted.
5. A list of abbreviations and acronyms should be added.
6. It is not a good idea that the authors refer a simple sentence to several references.
7. All equations need an appropriate reference.
8. The objectives of the work should be clearly highlighted.
9. What software was used in this study?
10. How the results were validated?

Author's Response to Decision Letter for (RSOS-201038.R0)

See Appendix A.

Decision letter (RSOS-201038.R1)

Dear Dr Mao:

Title: Theoretical analysis of superadiabatic combustion for non-stationary filtration combustion by excess enthalpy function

Manuscript ID: RSOS-201038.R1

It is a pleasure to accept your manuscript in its current form for publication in Royal Society Open Science. The chemistry content of Royal Society Open Science is published in collaboration with the Royal Society of Chemistry.

On behalf of the Subject Editor Professor Anthony Stace and the Associate Editor Dr Debashree Ghosh.

RSC Associate Editor
Comments to the Author:

(There are no comments.)

Reviewer(s)' Comments to Author:

Appendix A

Manuscript ID RSOS-201038

Title: Theoretical analysis of superadiabatic combustion for non-stationary filtration combustion by excess enthalpy function

Authors: Junrui Shi, Mingming Mao, Yongqi Liu and Jinsheng Lv

Article Type: Original Research Paper

Dear Editor and Reviewers,

We appreciate very much the helpful comments by the reviewers and the editor. The English is also carefully checked and polished. Following is the responses to all comments point by point. We numbered the comments and gave answers. All the revised parts or added content are distinguished by yellow base. Our responses on their questions and suggestions are as follows.

RSC Subject Editor:

Comments to the Author:

(There are no comments.)

Reviewers' Comments to Author:

Reviewer: 1

Comments to the Author(s)

The manuscript (RSOS-201038) presents a theoretical analysis of superadiabatic combustion for the non-stationary filtration combustion in porous medium. The excess enthalpy function (H equation) is derived based on the energy equations and

species conservation equation. An analysis of the terms in H equation is conducted and it is found that the thermal conductivity ratio between the solid and gas phases is a dominative factor among the four terms. It is an interesting research on the filtration combustion in porous medium which will help to understand the mechanism of the superadiabatic combustion. My specific comments are as follows,

1. More new and relevant literature should be cited and introduced in the introduction.

ANSWER: We accept this comment. New literature was added in the revised introduction part, please see line 17, page 5.

2. How about the grid of computing domain?

ANSWER: A uniform square grid with a size of 1 mm was used in the computation domain. We added comment on this issue, please see line 1, page 11.

3. The deviation in the Figure 5 is attributed to the different structure of porous media applied in the experiment. More discussions are needed to clarify the problem.

ANSWER: We accept this comment. please see line 8, page 14.

4. The conclusion discussion should be more detailed.

ANSWER: We accept this comment. please see the revised conclusion part.

5. It is recommended to carefully check the grammar and English language before resubmission. To list but a few as below:

ANSWER: We accept this comment. The grammar and English language were carefully checked and corrected point by point as below.

Line 14, page 9, “the thermal conductivity ratio between the solid and gas phases (Γ_s)

is” should be “the thermal conductivity ratio (Γ s) between the solid and gas phases is”.

ANSWER: We accept this comment. We have corrected the mistake.

Line 12, page 2, “using superadiabatic combustion” should be “from superadiabatic combustion”. “using superadiabatic combustion” has been replaced with “from superadiabatic combustion”.

ANSWER: We are sorry for the mistake. “using superadiabatic combustion” should be “from superadiabatic combustion”.

Line 7 page 3, “opposite directions” should be “opposite direction”.

ANSWER: We are sorry for the mistake. “opposite directions” has been replaced with “opposite direction”.

Line 9 page 3, “Equation. (1)” should be “Equation (1)”.

ANSWER: We are sorry for the mistake. “Equation. (1)” has been replaced with “Equation (1)”.

Line 17, page 5, “H equation are” should be “H equation is”.

ANSWER: We are sorry for the mistake. “H equation are” has been replaced with “H equation is”.

T0, Tn are confused in the manuscript.

ANSWER: We are sorry for the mistake. All “T_n” have been replaced with “T₀”.

Line 16, page 10, “is imposed” should be “are imposed”.

ANSWER: We are sorry for the mistake. “is imposed” has been replaced with “are imposed”.

Reviewer: 2

Comments to the Author(s)

1. English should be improved.

ANSWER: We accept this comment. The English is carefully checked and polished.

2. In the Abstract, you should clearly mention whether your paper is experimental or analytical and what software was used.

ANSWER: We accept this comment. We have added comment on this issue, “The superadiabatic combustion for non-stationary filtration combustion is analytically studied.”, “The governing equations with boundary conditions are solved by commercial software Fluent.”, please see the revised abstract part.

3. The most important quantitative results should be mentioned in the abstract.

ANSWER: We accept this comment. We added comment on this issue. “The maximum non-dimensional gas and solid temperatures in the flame zone are greater than 3 for equivalence ratio of 0.15.”, please see the revised abstract part.

4. The novelty of the paper should be highlighted.

ANSWER: We accept this comment. The novelty of the paper has been highlighted, please see line 13, page 5 and line 20, page 5.

5. A list of abbreviations and acronyms should be added.

ANSWER: We accept this comment. The main symbols were presented in Table 1 and thermodynamic properties and coefficient of reaction were summarized in Table 2.

The abbreviations and acronyms were explained in the manuscript.

6. It is not a good idea that the authors refer a simple sentence to several references.

ANSWER: Yes, in the introduction part we cited and classified the references, thus we cite several references in one sentence, we hope the understanding from the reviewer. For example, “The energy concentration phenomenon and combustion characteristics in porous burners have been investigated extensively in the literature including experimental [3-8], numerical [9-16] and analytical studies [16-27].” In this sentence, we cited several reference in one sentence, and we wanted to show that the energy concentration phenomenon has been studied by different methods.

7. All equations need an appropriate reference.

ANSWER: We accept this comment. In fact, the appropriate reference was cited, “Under the above assumptions, a set of differential equations can be obtained and expressed as follows [17]”, please see line 19, page 6.

8. The objectives of the work should be clearly highlighted.

ANSWER: We accept this comment. The objectives of the work have been highlighted, please see line 20, page 5. “In the following section, we formulate a mathematical model of filtration combustion in a packed bed and the H equation is theoretically derived based on a one-dimensional and two-temperature model. Subsequently, the source terms in the non-dimensional H function is examined and the effects of Γ_s and porosities on the superadiabatic combustion effect are discussed”.

9. What software was used in this study?

ANSWER: We accept this comment. We have added comments on this issue, “The H

equation is based on the solution of Eqs. (9), (10) and (11) using the finite-volume method by Fluent”, please see line 3, page 11.

10. How the results were validated?

ANSWER: The predictions were validated against experiment [3] in Fig. 5. Please see Fig. 5 and the discussion.

Sincerely yours

Junrui Shi and Mingming Mao